# Nurses' Work Environment during the COVID-19 Pandemic in a Person-Centred Practice—A Systematic Review

**Cicilia Nagel** [1,2,*] **, Albert Westergren** [1] **, Sophie Schön Persson** [1] **, Petra Nilsson Lindström** [1] **, Åsa Bringsén** [1] **and Kerstin Nilsson** [1,2,*]

1. Faculty of Health Sciences, Kristianstad University, 291 88 Kristianstad, Sweden;
albert.westergren@hkr.se (A.W.); sophie.schon@hkr.se (S.S.P.); petra.nilsson@hkr.se (P.N.L.);
asa.bringsen@hkr.se (Å.B.)
2. Faculty of Medicine, Lund University, 221 00 Lund, Sweden
* Correspondence: cicilia.nagel@med.lu.se (C.N.); kerstin.nilsson@med.lu.se (K.N.)

**Abstract:** The work environment and especially the psychosocial work environment influence the mental and physical well-being of employees. The aim of this study was to identify and analyse the state of knowledge regarding nurses' work situation, health, and person-centred work during the COVID-19 pandemic through a systematic review. Methods: Systematic Review, nine included articles. The theoretical swAge model was used as the framework in a deductive content analysis. Results: The result was presented in the nine determinate areas from the swAge model and showed that all nine determinate areas of the swAge model were of importance to both the nurses' sustainable work situation during the COVID-19 pandemic and to person-centred care. The COVID-19 pandemic has had a negative effect on nurses' health, both physically but especially psychologically, with high levels of depression, anxiety, and burnout. Nurses experienced a lack of control and support from organizations. They had to work with limited resources and sometimes care for patients beyond their expertise. Conclusion: There is a further need for more studies that address person-centredness from an organisational perspective with the intention to develop strategies and measure activities on how to make the nurses' work situation more sustainable, and to increase their ability to give more person-centred care.

**Keywords:** person-centred; organization; work environment; nurse; COVID-19

## 1. Introduction

In December 2019, the world became aware of a new coronavirus called SARS-CoV-2, and it became upgraded to a pandemic in March 2020, since it has had a profound effect around the world [1]. Healthcare workers, and primarily nurses, are on the frontline of this pandemic, where they are responsible for the care of patients. They had to perform their duties and face higher risks to their own health and risks of infection, which also gave exposure to hazards such as psychological distress, fatigue, and trauma [1]. The pandemic has been described as a gigantic strain experiment on health care staff and the health care organization [1–5]. According to the World Health Organization (WHO), there was a worldwide shortage of nurses, even before the onset of the COVID-19 pandemic; one factor that is causing this shortage is that some nurses leave the profession after only working for a few years [1–4]. The lack of nurses has resulted in an extreme challenge during the COVID-19 pandemic, to both nurses and other healthcare staff who worked during the pandemic, as well as to the healthcare organisations [5].

Work has a significant impact on people's health and healthy workplaces are beneficial not only for employees, but also for organisations and for society [6]. Decent work is also one of the United Nations' Sustainable Development Goals [7]. A healthy workplace is defined as one in which workers and managers work together to use a constant improvement process to safeguard and encourage the health, safety, and well-being of all workers

and the sustainability of the workplace [3]. A sustainable work situation for the employees is significant for a healthy organisation which attracts people to work, as well as better health, and thus also a higher possibility of employability with increased age [8–11]. An extremely stressful work situation, as has been the case in some parts of the health care system during the pandemic, there are shortcomings in the measures that needed to be taken. It is therefore important to detect shortcomings and problems in the work situation during the pandemic for nurses, and what needs to be improved to support healthy and sustainable employability. Areas of employability, and whether individuals can and want to work or not, have been stated as nine impact and determinant areas connected to a sustainable healthy working life [8–11], i.e., (1) the employees self-rated health and diagnoses; (2) physical work environment; (3) mental work environment; (4) working hours, work pace, possibility to time for recuperation; (5) personal financial situation; (6) personal, social environment outside work; (7) work social environment at the workplace; (8) stimulation, appreciation, and motivation within work tasks; (9) competence, skills, and possibility of acknowledging development in work. Regardless of a pandemic, it is the nurses' care responsibility to cherish the patient's dignity, integrity, and autonomy. It is therefore important to investigate how these nine impact and determinant areas have worked for nurses during the pandemic for increased knowledge and measures against staff shortages and future challenges in healthcare.

Healthcare is evolving towards becoming increasingly person-centred. Person-centredness is an approach to practice that can improve the possibility for nurses to fulfil their care responsibility through the formation and fostering of healthful relationships between all care providers, service users, and others significant to them in their lives [12–15]. A prerequisite for person-centeredness is to understand the individual as a unique person and places the unique individual as a subject in the centre, and not reduce them to an object of disease or symptom without distinguishing their individual conditions and personality. The person-centred perspective is underpinned by values of respect for persons (personhood), respect for the individual's rights to self-determination, building mutual trust and understanding, and treating the individual according to what strengthens his/her well-being [12,13]. Higher levels of person-centred care in the health and medical care organisation are stated as statistically significant, associated with less job strain, and higher staff satisfaction among nurses [16]. However, the ability for nurses to be person-centred in their work is influenced by the work culture of the organization [12–15]. It is enabled by cultures of empowerment that foster continuous approaches to practice development [12]. Additionally, the nurses need to be given organisational conditions to be able to provide person-centred care.

As the healthcare in most countries lately evolves towards more person-centred care, it is important that healthcare organizations have a mutual understanding of what it entails and requires in order for healthcare professionals to work according to person-centredness, even in a pandemic. Based on this, there is a need to study how person-centredness is viewed from an organisational perspective [12,13]. It is therefore important to study and map the state of knowledge regarding nurses' work situation and health during the COVID-19 pandemic, to investigate whether this affected their own situation and health, and to investigate whether this affected the nurses' ability to work in a person-centred manner during the COVID-19 pandemic.

The aim of this study was to identify and analyse the state of knowledge regarding nurses' work situation, health, and person-centred work during the COVID-19 pandemic through a systematically review. The aim was also to identify any knowledge gap regarding the nurse's work situation with importance to their own health and to better person-centred health and medical care.

## 2. Materials and Methods

### 2.1. Systematic Review

The purpose of a research review is to assess what has been studied already, and to enable people to plan what more needs to be studied [17]. In a systematic review, there is a general expectation that at least three databases will be used [18]. The method used in this investigation was a systematic literature review, performed in several steps in accordance with literature review guidelines in the Cochrane Handbook for Systematic Reviews [19]. The PRISMA guidelines were also read and adhered to [20], as well as the five steps to conducting a systematic review described by Khan et al. [21]. The first step was to frame the questions for the review, and the second was to identify relevant articles. First, a list of criteria was drawn up in relation to the aim to increase the validity and limit the search to the articles most relevant to the topic of this investigation.

The criteria for the inclusion of articles in the literature review (keywords are italicized below):

i.    empirical investigations
ii.   *nurse* was the interested profession for this systematic review, and therefore nurses had to be investigated
iii.  the nurses should have worked during the COVID-19 pandemic
iv.   the nurses *work environment* situation was the topic
v.    the *organisation* of the work situation and work environment was also of importance
vi.   due to the increased importance of person-centred the care, the word *person-centred* was important
vii.  the article needs to be a full text paper
viii. the published or accepted articles need to have been processed by peer-review
ix.   published in scientific journals
x.    in English language
xi.   as the authors of this paper is from Sweden paper published in Nordic languages was also stated to be possible for inclusion.

Exclusion criteria were:

i.    articles that did not apply to the aim of the study, i.e., nurses work situation during the COVID-19 pandemic
ii.   review papers or other publications that not in first line handle performed empirical investigations
iii.  literature that was not scientifically published
iv.   articles that were not available in full text
v.    articles published in another language than English or Nordic language.

The search was not limited to publication year, as SARS-CoV-2 (COVID-19) was not present before 2019. Literature searches were conducted in the following electronic databases: CINAHL, PubMed, Medline, and Scopus, by researchers CN and KN. Relevant articles were identified in several steps. First, a search for articles was conducted in each of the databases using the following five different keywords: person-centred, work environment, organisation, or organization (to include both English and American spelling), nurse, and COVID-19. The keywords were combined with Boolean operators (i.e., AND, and OR). While conducting our search, we found that by combining all the keywords with person-centred, no articles were found, thus forcing us to not use that particular keyword.

### 2.2. Analysis Method

The included articles were initially thoroughly assessed for quality by both researchers CN and KN, and thereafter analysed individually by researchers CN and KN through deductive content analysis. Subsequently, both authors compared and discussed their findings in order to sort the relevant parts of the collected data in the articles. Deduction can be said to establish a conclusion from the general to the individual. Knowledge and theories from previous research are the bases in a deductive content analysis to refine, and

possibly extend, a theoretical framework [22]. Deductive content analysis is a suitable choice when an existing theory involves the application of conceptual categories in the analysis of a new context. Content analysis can also follow an inductive approach which mean that any text that does not fit in the existing theory, or the predefined categories is given new codes in the content analysis to verify any new categories [23].

The swAge model was the theoretical model that was used as the framework in the deductive content analysis. The swage model consists of nine different determinant areas that are important to a sustainable working life for all ages and that relate to the four spheres of determination regarding employability, and the possibility of being able to and willing to be part of working life [8–11]. Those four spheres and nine determinant areas are:

➢ The health effects of the work environment, which include the following areas of determination:

    (1) Self-rated health, diagnoses, and diverse physical and mental health functionality in work,

    (2) physical work environment with unilateral movements, heavy lifting, risk of accidents, climate, chemical exposure, and risk of contagion,

    (3) mental work environment; stress and fatigue syndrome, threats, and violence

    (4) working hours, work pace, and possibility of recuperation during and between work shifts.

Adequate health is a prerequisite for employability and to be included in working life [8–11]. Work life affects biological ageing, mental and physical health, and the need for recovery based on physical and mental stresses, but also by the strengthening impacts of our work.

➢ Financial incentives are associated with society's control of various financial carrots and sticks, such as through the social insurance system. Financial incentives include the following determinant area:

    (5) The personal financial situation's effects on individuals' needs and willingness to work.

Issues with employability due to ill health, lack of skills and lack of support risk causing exclusion from working life and a poorer financial situation for the individual e.g., through sick leave, unemployment, and early retirement, not least in tough times [8–11].

The organisations and workplace finances rule the staffing ratio, which equipment and techniques that can be used to facilitate a more sustainable work environment thus increasing long time employability.

➢ Relationships, social support, and participation, i.e., attitudes in the social context in which the individual finds himself/herself, whether the individual feels included or excluded in the group and receives sufficient social support from the environment when needed, include the areas of determination:

    (6) The effects of the personal social environment, with family, friends, and leisure context, and

    (7) the social work environment with leadership, discrimination, and the significance of the employment relationship context for individuals' work.

Every employee has a personal life and a social environment, and aspects in their personal relationships can affect the individual' opportunities and willingness to work [8–11].

➢ Execution of tasks and activities, relate to individual and instrumental support and include the following areas of determination:

    (8) Motivation, appreciation, satisfaction, and stimulation in work tasks, and

    (9) knowledge, competence, and the importance of competence development for the individual's work.

Working life is constantly evolving and the employability of an individual depends on their ability to meet the requirements of knowledge and skills in order to perform the activities and tasks that their work entails [8–11]. The tasks and activities at work can be a source of motivation, stimulation, and joy, but can also be a source of monotony, dissatisfaction, and inactivity.

The analysis of the included articles started by creating a formative categorisation matrix with four pre-set categories, in this article named spheres, based on the theory and the determinant spheres in the swage model [8–11].

The overview of each article is listed in Table 1 and the analysis result is described in Section 3.3 below.

Table 1. Result overview of articles and their content related to the nine impact and determinant areas in the swAge model.

| Author, Year, Reference, Article Number | Aim | Method, Participants | Results | Discussion and Conclusion | The Nine Determinant Areas Important for a Sustainable Working Life (swAge-Model): 1. Diagnosis and Self-Rated Health 2. Physical Work Environment 3. Mental Work Environment 4. Workhours, Pace, Recovery 5. Economics 6. Private-Social Environment 7. Work-Social Environment 8. Motivation, Stimulance, Task Satisfaction 9. Knowledge, Competence |
|---|---|---|---|---|---|
| Da Rosa et al., 2021. Factors associated with nurses' emotional distress during the COVID-19 pandemic [24]. | To examine the prevalence of emotional distress and the associated factors among nurses practicing in South Dakota during the COVID-19 pandemic. | Quantitative design. An online survey. Respondents: 1505 licenced nurses in South Dakota during the pandemic Emotional distress was measured using the Depression, Anxiety, and Stress Scale (DASS-21). (July–August 2020). | Overall emotional distress was reported by 22.2%, while anxiety, depression, and stress were 15.8%, 14.5%, and 11.9% respectively. Factors associated with moderate to severe emotional distress, depression, anxiety, and stress were as follows: concerns for worsening of pre-existing mental health condition, job dissatisfaction, encountering higher number of COVID-19 cases at one's work facility, feeling unprepared for the pandemic, and concern for contracting the illness (all $p < 0.05$). | A high prevalence of emotional distress among nurses highlights the factors associated with emotional distress during the COVID-19 pandemic. Promoting appropriate support is imperative to reduce nurses' emotional distress and promote psychological well-being during the COVID-19 world health crisis and in future pandemics. | 1,5,6,7,8,9 |

**Table 1.** *Cont.*

| Author, Year, Reference, Article Number | Aim | Method, Participants | Results | Discussion and Conclusion | The Nine Determinant Areas Important for a Sustainable Working Life (swAge-Model): 1. Diagnosis and Self-Rated Health 2. Physical Work Environment 3. Mental Work Environment 4. Workhours, Pace, Recovery 5. Economics 6. Private-Social Environment 7. Work-Social Environment 8. Motivation, Stimulance, Task Satisfaction 9. Knowledge, Competence |
|---|---|---|---|---|---|
| Bergman et al., 2021. Registered nurses' experiences of working in the intensive care unit during the COVID-19 pandemic [25]. | To describe Swedish registered nurses' experiences of caring for patients with COVID-19 in ICUs during the pandemic. | Mixed method survey design. 282 respondents. An online questionnaire was distributed through social media to registered nurses who had been working in the ICU during the COVID-19 outbreak. Data were collected for 1 week (May 2020). | Of the 282 nurses who participated, the majority were ICU nurses ($n = 151$; 54%). Among non-intensive care nurses, only 19% received introduction to the COVID-19 ICU ($n = 26$). Three categories: tumbling into chaos, diminished nursing care, and transition into pandemic ICU care. Participants perceived that patient safety and care quality were compromised, and that nursing care was severely deprioritized during the pandemic. Not being able to to provide nursing care resulted in ethical stress. An increased workload and worsened work environment affected nurses' health and well-being. | Nurses perceived that patient safety and quality of care were compromises during the pandemic. This resulted in ethical stress among nurses, which may have affected their physical and psychosocial well-being. | 1,2,3,4,5,6,7,8,9 |

**Table 1.** *Cont.*

| Author, Year, Reference, Article Number | Aim | Method, Participants | Results | Discussion and Conclusion | The Nine Determinant Areas Important for a Sustainable Working Life (swAge-Model): 1. Diagnosis and Self-Rated Health 2. Physical Work Environment 3. Mental Work Environment 4. Workhours, Pace, Recovery 5. Economics 6. Private-Social Environment 7. Work-Social Environment 8. Motivation, Stimulance, Task Satisfaction 9. Knowledge, Competence |
|---|---|---|---|---|---|
| George et al., 2021. Roles and experiences of registered nurses on labor and delivery units in the United States during the COVID-19 pandemic [26]. | To examine the roles and experiences of labor and delivery nurses (LD) during the COVID-19 pandemic. | Mixed method design. Quantitative data from a cross-sectional online national survey. Qualitative data was an open-ended question about changes to their roles during the COVID-19 pandemic. Respondents: 757 nurses (July–August 2020). | Four major categories emerged: Changes in roles and responsibilities; Adaptions to changes; Psychological changes; and Perceived effects on labor support. Nearly half ($n = 328$) of respondents reported changes in their roles and responsibilities during the pandemic. Infection control policies and practises along with the stress of rapidly changing work environment affected the provision of labor support and personal well-being. | Policies and practises that can fascilitate the ability of LD nurses to safely and securely remain at the bedside and provide high-touch, hands-on labor support are needed. | 1,2,3,4,5,6,8 |

**Table 1.** *Cont.*

| Author, Year, Reference, Article Number | Aim | Method, Participants | Results | Discussion and Conclusion | The Nine Determinant Areas Important for a Sustainable Working Life (swAge-Model): 1. Diagnosis and Self-Rated Health 2. Physical Work Environment 3. Mental Work Environment 4. Workhours, Pace, Recovery 5. Economics 6. Private-Social Environment 7. Work-Social Environment 8. Motivation, Stimulance, Task Satisfaction 9. Knowledge, Competence |
|---|---|---|---|---|---|
| Gago-Valiente et al., 2021. Emotional exhaustion, depersonalization, and mental health in nurses from Huelva: a cross-cutting study during the SARS-CoV-2 pandemic [27]. | To examine the prevalence of emotional exhaustion, depersonalization, and possible non-psychotic psychiatric disorders in nursing professionals during the COVID-19 pandemic. | Quantitative design. Descriptive cross-sectional study. Respondents: 318 nursing professionals (April–June 2020). | Nurses who had contact with SARS-CoV-2 in their work environment showed higher levels of emotional exhaustion (49.6%) and depersonalization (34.3%) than nurses who had no contact (38.3% and 21.1% respektively). Among the cases of emotional exhaustion, there were around 60% with non-psychotic psychiatric symptoms compared to 28.5% who did not show it. On the other hand, in the cases of depersonalization, almost 40% evidenced non-psychotic psychiatric symptoms, compared to 25% who did not. | The nursing staff who have had contact with COVID-19 in their work environment had poorer state of health leading to high emotional exhaustion, high depersonalization, with a likely precense of a non-psychotic psychiatric pathologies. In this study sample the men, in general, showed a poorer state of mental health than that of the women. | 1,3,4,6,9 |

**Table 1.** *Cont.*

| Author, Year, Reference, Article Number | Aim | Method, Participants | Results | Discussion and Conclusion | The Nine Determinant Areas Important for a Sustainable Working Life (swAge-Model): 1. Diagnosis and Self-Rated Health 2. Physical Work Environment 3. Mental Work Environment 4. Workhours, Pace, Recovery 5. Economics 6. Private-Social Environment 7. Work-Social Environment 8. Motivation, Stimulance, Task Satisfaction 9. Knowledge, Competence |
|---|---|---|---|---|---|
| Firew et al., 2020. Protecting the front line: a cross-sectional survey analysis of the occupational factors contributing to healthcare workers' infections and psychological distress during the COVID-19 pandemic in the USA [28]. | To investigate factors contributing to healthcare workers (HCWs) infection and psychological distress for HCWs, with COVID-19 exposure risk during the COVID-19 pandemic in the USA. | Quantitative design. A cross sectional survey of HCWs Respondents: 2040 (physicians 31%, nurses 27%, emergency medical technicians (EMTs) 13%, non-clinical staff 29%) from 48 states, the District of Columbia, and US territories (May 2020) | HCWs in the emergency department (31.64%) were more likely to contract COVID-19 compared with HCWs in the ICU (23.17%) and inpatient settings (25.53%). HCWs that contracted COVID-19 reported higher levels of depressive symptoms (mean diff. = 0.31; 95% CI 0.16 to 0.47), anxiety symptoms (mean diff. = 0.34; 95% CI 0.17 to 0.52) and burn-out (mean diff. = 0.54; 95% CI 0.36 to 0.71). Primary outcome: prevalence of self-reported COVID-19 infection, in addition to burn-out, depression and anxiety symptoms. | HCWs have experienced significant physical and psychological risk while working during the COVID-19 pandemic. These findings highlight the urgent need for increased support for providing physical and mental health well-being. | 1,2,3,5,6,7 |

**Table 1.** *Cont.*

| Author, Year, Reference, Article Number | Aim | Method, Participants | Results | Discussion and Conclusion | The Nine Determinant Areas Important for a Sustainable Working Life (swAge-Model): 1. Diagnosis and Self-Rated Health 2. Physical Work Environment 3. Mental Work Environment 4. Workhours, Pace, Recovery 5. Economics 6. Private-Social Environment 7. Work-Social Environment 8. Motivation, Stimulance, Task Satisfaction 9. Knowledge, Competence |
|---|---|---|---|---|---|
| Galanis et al., 2021. Fear of COVID-19 among nurses in mobile COVID-19 testing units in Greece [29]. | To assess the level of fear of COVID-19 among nurses in mobile COVID-19 testing units and compare it with demographic characteristics | Quantitative design. A cross-sectional study Respondents: 57 nurses working in mobile testing units. The fear of COVID-19 scale was used to measure fear of the COVID-19 pandemic. (November–December 2020) | Among nurses, 31.6% experienced elevated fear of COVID-19 indicative of presence of anxiety and post-traumatic stress symptommatology were 22.8% and 17.5%. Fear of COVID-19 was not affected by demographic variables. Fear was higher in females, nurses who had children, and nurses who lived with others. Increased clinical experience was related with decreased fear. | A secure work environment with access to personal protective equipment (PPE) and relevant training for nurses in these units could decrease fear of COVID-19 and increase work performance. | 1,3,6,9 |

**Table 1.** *Cont.*

| Author, Year, Reference, Article Number | Aim | Method, Participants | Results | Discussion and Conclusion | The Nine Determinant Areas Important for a Sustainable Working Life (swAge-Model): 1. Diagnosis and Self-Rated Health 2. Physical Work Environment 3. Mental Work Environment 4. Workhours, Pace, Recovery 5. Economics 6. Private-Social Environment 7. Work-Social Environment 8. Motivation, Stimulance, Task Satisfaction 9. Knowledge, Competence |
|---|---|---|---|---|---|
| Suryavanshi et al., 2020. Mental health and quality of life among healthcare professionals during the COVID-19 pandemic in India [30]. | To assess the mental health and quality of life (QoL) of Indian Health Care Professionals (HCPs), the fourth highest-burden country for COVID-19. | Quantitative design. Online survey) Respondents: 197 (24% nurses, 34% physichans, 42% other health care workes) (May 2020) | Of 197 HCPs assessed, 130 participants (66%) worked in public hositals, 47 (24%) were nurses, 66 (34%) physicians. A large proportion reported symptoms of depression (92.47%), anxiety (98.5%), and low QoL (89.45%). Odds of combined depression and anxiety were 2.37 times higher among single HCPs compared to married (95% CI: 1.03–4.96). Work environment stressors were associated with 46% increased risk of combined depression and anxiety (95% CI: 1.15–1.85). Moderate to severe depression and anxiety were independently associated with increased low QoL OR: 3:19 (95% CI: 1.30–7.84), OR:2.84 (95% CI: 1.29–6.29). | A high prevalence of symptoms of depression and anxiety and low QoL among Indian HCPs during the COVID-19 pandemic. There is an urgent need to prevent and treat mental health symptoms among frontline HCPs. | 1,2,4,5,6,7,9 |

**Table 1.** *Cont.*

| Author, Year, Reference, Article Number | Aim | Method, Participants | Results | Discussion and Conclusion | The Nine Determinant Areas Important for a Sustainable Working Life (swAge-Model): 1. Diagnosis and Self-Rated Health 2. Physical Work Environment 3. Mental Work Environment 4. Workhours, Pace, Recovery 5. Economics 6. Private-Social Environment 7. Work-Social Environment 8. Motivation, Stimulance, Task Satisfaction 9. Knowledge, Competence |
|---|---|---|---|---|---|
| Norful et al., 2021. Primary drivers and psychological manifestation of stress in frontline healthcare workforce during the initial COVID-19 outbreak in the United States [31]. | To understand the physical and psychological impact of high stress clinical environments and contributory factors of burnout in multi-disciplinary healthcare workforce during the initial outbreak of COVID-19. | Qualitative design. In-person interviews Respondents: 55 healthcare workers (21 registrerd nurses, 5 respiratory therapists, 12 physitians, 4 pharmacists, 13 patent care technicians) (March–April 2020) | Themes revolved around three main areas: fear of uncertainty, physical and psychological manifestations of stress, and resilience building. Shifting information, a lack of PPE, and fear of infecting others prompted worry for those working with COVID-infected patients. Stress manifested more psychologically than physically. Individualized stress mitigation efforts, social media and organizational transparence were reported by healthcare workers to be effective against rising stressors. | In order to understanding stressors and supporting clinicians during healthcare emergencies, more research is necessary to effectively promote healthcare workforce well-being. | 1,3,4,5,6,7,9 |

Table 1. *Cont.*

| Author, Year, Reference, Article Number | Aim | Method, Participants | Results | Discussion and Conclusion | The Nine Determinant Areas Important for a Sustainable Working Life (swAge-Model): 1. Diagnosis and Self-Rated Health 2. Physical Work Environment 3. Mental Work Environment 4. Workhours, Pace, Recovery 5. Economics 6. Private-Social Environment 7. Work-Social Environment 8. Motivation, Stimulance, Task Satisfaction 9. Knowledge, Competence |
|---|---|---|---|---|---|
| Giusino et al., 2021. "We all held our own" Job demands and resources at Individual, Leader, Group and Organizational levels during COVID-19 outbreak in healthcare [32]. | To explore the fitness of integrating the Job Demands-Resources model (JD-R) and the Individual-Group-Leader-Organization (IGLO) framework to investigate the pandemic's impact on healthcare workers' mental health | Qualitative design. Semi-structured interviews and focus group interviews. Respondents: 21 senior and middle nurse managers and healthcare workers from the Department of Emergency, Department of Medicine, and Research Institute of Neuroscience. (September–October 2020) | Several COVID-19-related job demands and resources were found at all IGLO levels. Individual-level demands included emotional load, while resources included resilience and motivation. Group-level demand included social distancing, while resources included team support and cohesion. Leader-level demands included managers' workload, while resources included leader support. Organizational-level demands included work reorganization while resources included mental health initiatives | Integrating JD-R and IGLO proved feasible as job demands and resources could be categorized according to the different levels of the framework. The findings fills the lack of knowledge on how job demands and resources might unfold at different workplace levels during a pandemic. Results provide unitlevel evidence for designing and implementing multilevel interventions to manage healthcare workers' mental health during COVID-19 and future pandemics. | 1,2,3,4,6,7,8,9 |

**Table 1.** *Cont.*

| Author, Year, Reference, Article Number | Aim | Method, Participants | Results | Discussion and Conclusion | The Nine Determinant Areas Important for a Sustainable Working Life (swAge-Model): 1. Diagnosis and Self-Rated Health 2. Physical Work Environment 3. Mental Work Environment 4. Workhours, Pace, Recovery 5. Economics 6. Private-Social Environment 7. Work-Social Environment 8. Motivation, Stimulance, Task Satisfaction 9. Knowledge, Competence |
|---|---|---|---|---|---|
| van der Goot et al., 2021. Psychological distress among frontline workers during the COVID-19 pandemic: a mixed-methods study [33]. | To investigate experienced psychological distress during the COVID-19 pandemic from a self-determination theory perspective | Mixed-method design. Quantitative data: repeated measures: survey. Qualitative data: audio diaries. Respondents: 46 (50% nurses, 33% junior doctors, 17% hospital consults). (April–November 2020) | Quantitative results indicated that perceived psychological distress during COVID-19 was higher than pre-COVID and fluctuated over time. Need frustration, specifically autonomy and competence, was positively associated with psychological distress while need satisfaction, especially relatedness, was negatively associated with psychological distress. The qualitative thematic analysis stated that especially organizational logistics frustrated autonomy, and unfamiliarity with COVID-19 frustrated competence. Despite many need frustrating experiences, a strong connection with colleagues and patients were important sources of relatedness support that seemed to mitigate psychological distress | Challenging times require healthcare organisations to better support their professionals by tailored formal and informal support. The authors propose to address both indirect (e.g., organisation) and direct (e.g., colleagues) elements of the clinical and social environment in order to reduce need frustration and enhance need satisfaction. | 1,3,6,7,8,9 |

**Table 1.** *Cont.*

| Author, Year, Reference, Article Number | Aim | Method, Participants | Results | Discussion and Conclusion | The Nine Determinant Areas Important for a Sustainable Working Life (swAge-Model): 1. Diagnosis and Self-Rated Health 2. Physical Work Environment 3. Mental Work Environment 4. Workhours, Pace, Recovery 5. Economics 6. Private-Social Environment 7. Work-Social Environment 8. Motivation, Stimulance, Task Satisfaction 9. Knowledge, Competence |
|---|---|---|---|---|---|
| Thomas et al., 2021. COVID-19 and moral distress: a pediatric critical care study [34]. | To investigate whether pediatric critical care professionals are experiencing moral distress during the COVID-19 pandemic and, if so, for what reasons. | Quantitative design. An exploratory survey. Respondents: 337 paediatric critical care professionals 26% nurses, 49% physichans, 16% respiratory therapists, other health care workes 9%, via the Pediatric Acute Lung Injury and Sepsis Investigators Network from (April–May 2020) | Overall, 85.8% of survey respondents reported moral distress. Nurses reported higher degrees of moral distress than other professional groups. Inducers of moral distress were related to challenges to professional integrity and lack of organizational support. 5 themes: psychological safety, expectations of leadership, connectedness through a moral community, professional challenges, and professional vs. social responsibility. Most respondents were confident in their ability to reason through ethical dilemmas (76%) and think clearly when confronting an ethical challenge even when pressured (78.9%). | During the COVID-19 pandemic, pediatric critical care professionals are experiencing moral distress due to various factors that challenge their professional integrity. Despite these challenges, they also exhibit attributes of moral resilience. Organizations have opportunities to cultivate a psychologically safe and healthy work environment to mitigate anticipatory, present, and lingering moral distress. | 1,2,3,4,5,6,7,8,9 |

**Table 1.** *Cont.*

| Author, Year, Reference, Article Number | Aim | Method, Participants | Results | Discussion and Conclusion | The Nine Determinant Areas Important for a Sustainable Working Life (swAge-Model): 1. Diagnosis and Self-Rated Health 2. Physical Work Environment 3. Mental Work Environment 4. Workhours, Pace, Recovery 5. Economics 6. Private-Social Environment 7. Work-Social Environment 8. Motivation, Stimulance, Task Satisfaction 9. Knowledge, Competence |
|---|---|---|---|---|---|
| Sheppard et al., 2021. Nursing moral distress and intent to leave employment during the COVID-19 pandemic [35]. | To explore the levels of moral distress (MD) among registered nurses in the practice environment and inform the nurse leaders about the impact of MD on nursing turnover. | Quantitative design. Questionnaire. Respondents: 129 registered nurses. Survey Measure of Moral Distress for Healthcare Professionals (MMD-HP) was used. (July–August 2020) | T-tests showed significant differences for 16 of 27 MMD-HP items in registered nurses intent to leave. RNs had 2.9 times the odds of intent to leave ($p = 0.019$) due to perceived issues with patient quality and safety and 9.1 times the odds of intent to leave ($p < 0.001$) due to perceived issues with the work environment. Results explained 40.3% of outcome variance. | MD related to work environment or patient quality and safety were significant factors in registered nurses intent to leave their positions. | 1,3,4,5,7,9 |

**Table 1.** *Cont.*

| Author, Year, Reference, Article Number | Aim | Method, Participants | Results | Discussion and Conclusion | The Nine Determinant Areas Important for a Sustainable Working Life (swAge-Model): 1. Diagnosis and Self-Rated Health 2. Physical Work Environment 3. Mental Work Environment 4. Workhours, Pace, Recovery 5. Economics 6. Private-Social Environment 7. Work-Social Environment 8. Motivation, Stimulance, Task Satisfaction 9. Knowledge, Competence |
|---|---|---|---|---|---|
| Guedes dos Santos et al., 2021. Work environment of hospital nurses during the COVID-19 pandemic in Brazil [36]. | To investigate the nurses' work environment in university hospitals during the COVID-19 pandemic in Brazil | Mixed methods design. Quantitative data was collected by an online questionnaire. Qualitative data was collected through an open question Repondents: 104 nurses from three university hospitals. (April–June 2020. | The quantitative results showed that the responses to 'I received training on the correct use of personal protective equipment (PPE)' and 'I am afraid of being infected' had the best and worst evaluations respectively. The qualitative findings revealed five themes: feeling of insecurity, lack of PPE, lack of diagnostic tests, changes in the care flow and fear of the unknown. | The study highlighted the challenges faced by hospital nurses while caring for patients with COVID-19. | 1,2,3,4,5,7,9 |
| Diomidous, M., 2020. Sleep and motion disorders of physicians and nurses working in hospitals facing the pandemic of COVID-19 [37]. | To investigate the relationship between the physical activity and sleep disorders among healthcare professionals, particularly among medical doctors and nurses | Quantitative design. Questionnaire Respondents: 204 (102 medical doctors and 102 nurses). (February–April 2020). | The results of the statistical analysis showed that there are positive correlations between the level of physical activity during the daily work and the free time of the participants with parameters that are related to sleep disorders. | A stressful situation such as the COVID-19 pandemic can provide useful information in order to better understand the relationship between physical activity and sleeping disorders in similar working conditions | 1,4,9 |

**Table 1.** *Cont.*

| Author, Year, Reference, Article Number | Aim | Method, Participants | Results | Discussion and Conclusion | The Nine Determinant Areas Important for a Sustainable Working Life (swAge-Model): 1. Diagnosis and Self-Rated Health 2. Physical Work Environment 3. Mental Work Environment 4. Workhours, Pace, Recovery 5. Economics 6. Private-Social Environment 7. Work-Social Environment 8. Motivation, Stimulance, Task Satisfaction 9. Knowledge, Competence |
|---|---|---|---|---|---|
| Jo et al., 2021 Nurses' resilience in the face of coronavirus (COVID-19): An international view [38]. | To examine factors associated with nurses' resilience during the COVID-19 pandemic | Quantitative design. Cross-sectional descriptive study. Respondents: 904 nurses in Japan, Korea, Turkey, and the United States. (July–November 2020) | Fear of becoming infected, intention to leave nursing, and having had a positive COVID-19 test were negatively associated with resilience ($p < 0.05$). Regression analysis indicated that U.S. nurses had significantly higher resilience than nurses in the other countries examined ($p < 0.001$). Nurses reporting organizational support and those who participated in policy and procedure development had higher resilience score ($p < 0.01$). | Organizational support, involving nurses in policy development, and country of practice were found to be important resilience factors. The authors recommend to further determine the optimal practice environment to support nurse resilience. | 1,3,5,6,9 |

## 3. Results

### 3.1. Inclusion and Exclusion Result

When combining the keywords Organization/Organisation and Work environment, 13,650 articles were found. After adding the keyword Nurse to these keywords, 4431 articles were found. In the next step, 326 articles were identified, after combining the keywords Organization/Organisation; Work environment; Nurse; COVID-19 (Figure 1). After additionally including the last keyword Person-centred with the other keywords, no articles were found. To identify the reason for this lack of articles, a specific search was performed with only the keyword Person-centred, and 6325 articles were shown. However, most articles did not meet the criteria set and had no relevance for this systematic literature review. Most articles investigated patients' experience and not the nurses' experience of their work situation, and no one investigated the nurses' experiences during the COVID-19 pandemic. Due to this, the keyword person-centred was not used in the final search.

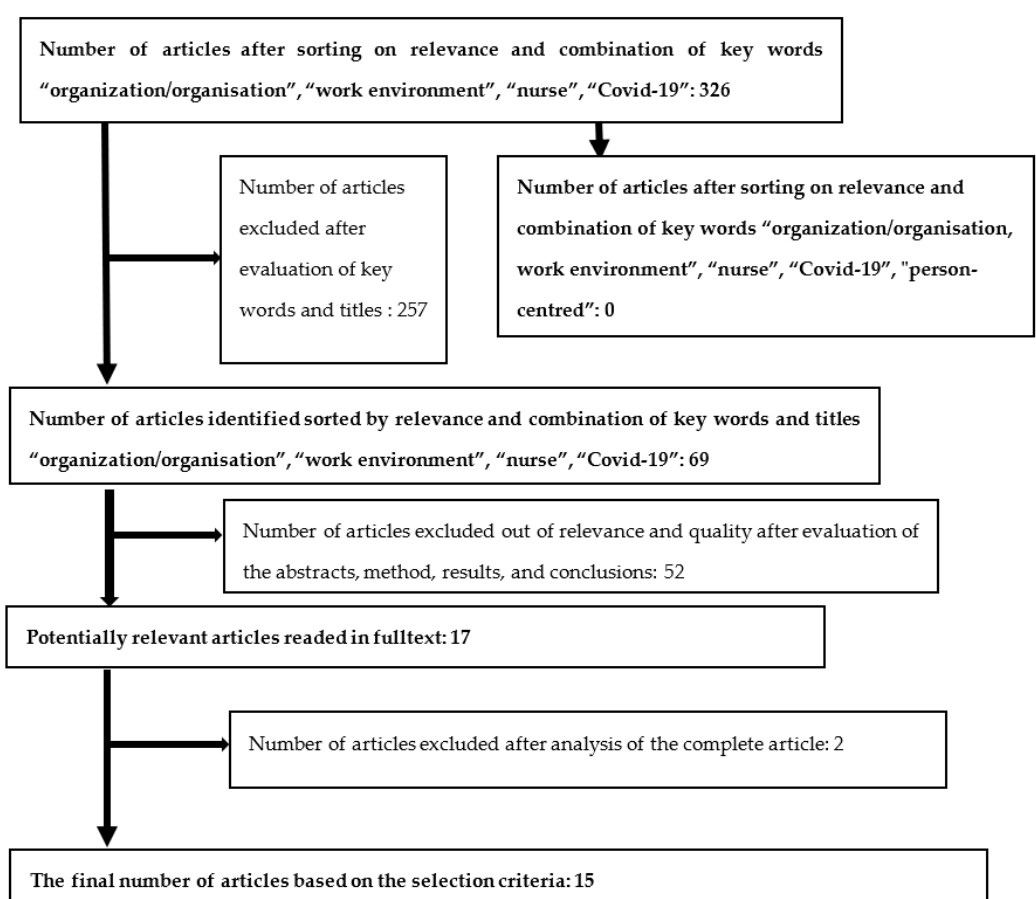

**Figure 1.** Selection process for relevant articles.

The final combination of keywords resulted in 326 remaining articles. However, even this final combination of keywords led to some articles that did not have any relevance. For example, the keyword "COVID-19" led to many articles about the virus itself. This systematic literature review aimed to focus on the nurses' situation in their work environment, and those articles were therefore sorted out. By reading through titles and keywords, 69 articles were considered interesting for the study. No articles in the Nordic language were identified. The 69 references were uploaded in the Rayyan software (www.rayyan.ai, (accessed on 10 November 2021)) where the articles once more were sorted by the keywords, and the aim and the abstracts were read through, and the quality of the studies was assessed by researchers CN and KN, which result in the exclusion of 52 articles that did not answer the aim of this study. The remaining 17 articles in the full text were then carefully read by

both researchers individually, and 2 articles were sorted out since these were not relevant to the aim. The remaining 15 articles were carefully read in the full text furthermore, and a resulting overview of the articles was created (Table 1). The article results were analysed and put into four pre-set spheres based on the theory and the determinant areas in the swage model (see Section 2.2).

*3.2. Result Overview of the Articles*

The COVID-19 pandemic is stated by the WHO to have started in 2020, therefore there were no articles before 2019, and the included articles were published between 2020 and 2021. The included articles in this review used different study designs. Of the articles two used a qualitative study design [30,31], nine used a quantitative study design [24,27–30,34,35,37,38] and four used a mixed method study design [25,26,33,36]. The investigations were performed with nurses in the countries of Sweden [25], U.S. [24, 26,28,31,34,35,38], Greece [29,37], Japan [38], Korea [38], Turkey [38], Italy [32], Spain [27], Brazil [36], India [30], and the Netherlands [33]. Together, the included articles covered all the determinate areas of a sustainable working life (swage model). Most of the articles handled the determinant areas: Diagnosis and self-rated health, Mental work environment and Economics.

An important finding was that when the key word "person-centred" was added to the other search terms, no articles were found at any of the four databases. However, all studies address factors that are important for subjecting and person-centring the nurse as a unique employee in the organisation of the work situation and the work environment, and of importance for person-centredness and subjecting the patients in healthcare. Therefore, the investigation of person-centredness in the nurses' work environment and in the organisation of care during the COVID-19 pandemic was analysed out of the factors in the definition of person-centred, and not out of the use of the term person-centred.

*3.3. Analyses of the Included Articles*

The analysis of the included articles began with the construction of a formative categorisation matrix with four pre-set categories based on the theory and the nine determinant areas in the swAge model [8–11].

3.3.1. Determinant Sphere: A: The Health Effects of the Work Environment

(1)    the employees own self-rated health and diagnoses:

During the first months of the pandemic nurses experienced fatigue [24,25]. The constant vigilance was described as draining [31]. Fear of becoming infected with COVID-19 was described by many and this fear caused anxiety [24,26,30,32,38]. Symptoms of stress and burnout, such as nightmares [25,37], having difficulty to breathe, feeling frustrated or scared were reported [26]. Many who had contact with COVID-19 in their work environment showed elevated levels of emotional exhaustion [25,27,31]. According to a study made in the U.S. [28], a higher percentage of work hours spent in close contact with COVID-19 patients was associated with higher levels of depression, anxiety, and burnout. A study made in Greece [29] showed that a considerable percentage of nurses developed psychological symptoms, in particular feelings of sorrow, anxiety, health anxiety, depression, and posttraumatic stress symptoms. This was also reported in a study made in India [30] where 90% of the study population reported symptoms of depression, anxiety, and low quality of life. Other participants also reported prevalence of depression and anxiety [24,27,29–31,35,36]. Isolation was associated with significantly higher levels of depressive symptoms [28]. Participants in several studies described feeling stressed [24,25,27,34]. Certain stressors were the complexity of symptoms of patients with COVID-19 [31], feeling unprepared for the pandemic [24], and not having access to COVID-19 tests [26,36]. A substantial number of participants experienced increased psychological distress during the COVID-19 pandemic [33]. Younger nurses were significantly more likely to report higher levels of stress [24]. A study from the U.S. [31] described how participants reported that stress

manifested more psychologically than physically although there were reports of physical implications such as skin irritation from long term protective mask use. Various degrees of physical symptoms such as headache, muscle- and joint aches, fever/chills, sore throat, coughing, nausea/vomiting, shortness of breath, loss/change of taste and smell were reported [28]. Isolation and living alone were two factors that were associated with significantly higher levels of depressive symptoms [28,30]. Concerns for worsening of pre-existing mental health conditions, job dissatisfaction and concerns for contracting the illness were factors associated with moderate to severe emotional distress [24]. Healthcare workers who were diagnosed with COVID-19 reported higher levels of depressive symptoms, anxiety symptoms, and burnout [28].

(2)     physical work environment at the workplace:

During the first stage of the pandemic, post-operative units or operating theatres were quickly redesigned to become Intensive Care Unit (ICU) wards to care for patients in need of mechanical ventilation [25]. Stressors primarily contributing to affecting the work environment were lack of knowledge [30] and lack of manpower [26,30]. Social distancing was described as complicated due to the lack of proper workspaces [32]. Participants stated that social distancing in the workplace was difficult to maintain, partly as it disrupted clinician-patient relationships [34] and partly since it was difficult when trying to support the labouring woman who could not always keep her mask on during labour [26]. A study from India [30] stated that there was a lack of adequate isolation wards for patients with COVID-19 while some participants in a study from Brazil [36] stated that the physical work environment was inadequate and sometimes unsafe for them to perform their duties in. Providing care to a large number of patients with COVID-19 as well as prolonged contact during work hours were associated with increased risk of infection [28]. An increased workload and worsened work environment affected nurses' health and well-being [25]. It was suggested that healthcare professionals' anxiety could be reduced by reducing patient load, having proper training in COVID-19 management guidelines, having adequate isolation wards and a sufficient supply of personal protective equipment [30].

(3)     mental work environment at the workplace:

A study from Spain [27] showed significant differences in emotional exhaustion and depersonalization, both when it came to age with an increase in age that was correlated with increased emotional exhaustion and depersonalization but also when it came to gender differences with women showing lower levels of suffering than men. Galanis et al. [29] reported that older nurses displayed higher levels of anxiety. Many of the articles stated increased mental demand in work environments and loss of control among nurses during the COVID-19 pandemic. Something the respondents in the different studies agreed on was a lack of control in some way. Whether it was that they felt that they only had little or even no input on their working condition, which generated stress [25] or that those who transferred to other units were required to join unfamiliar teams [26,33]. Not being able to influence scheduling or that rosters only were available at the last minute made the nurses feel a lack of control [32,33]. Many respondents perceived that a loss of control could challenge their professional integrity [34]. Some participants described how they just had to accept changes since they stood lower in the overall hierarchy [33]. There were reports of a wish from healthcare workers for more instruction about the constant organizational changes, which could have added some feeling of control over the situation [32]. Not being enough involved in decision making or feeling incompletely informed was reported by many participants, this sometimes led to some degree of uncertainty or frustration for instance when it was unclear if or when participants would return to their own speciality to provide care for non-COVID-19 patients [33]. Almost one third of participants in a study from Brazil [36] stated that they had a negative feeling/experience of working/coming to work in the care of patients with COVID-19. A few nurses expressed that they felt like no one cared about their or their family's life [31]. Nurses were exposed to highly stressful situations such as witnessing patients deteriorate without them being able to do

anything [33], having co-workers contract/being admitted to hospital with COVID-19 [28] or witnessing patients dying [32,34]. There were, however, incidences were patients improved, thus giving nurses a sense of hope for other patients and the strength to keep working [25].

Nurses that have worked during the COVID-19 pandemic have stated that they were considering leaving the profession due to additional and shifting job responsibilities, greater nurse-to-patient staffing ratios, and concerns for reduced standards of care [26,35,38] as well as a lack of personal protective equipment (PPE) and lack of support [33]. Moral distress was reported when nurses felt pressured by family and/or physicians to carry out what they believed to be aggressive or unnecessary treatments [34–36]. Nurses that perceived their work environment as morally distressing were nine times more likely to leave their positions [35]. Nurses in a study by George et al. [26] felt like they had to function as police officers and check the temperature/signs of COVID-19 on patients and support persons on arrival to the hospitals as well as enforcing mask regulations.

(4)     working hours, work pace, time for recuperation:

The COVID-19 pandemic has forced healthcare workers, including nurses, to work longer hours [25,32,36] and with greater workload [25,30–32,35,36]. Several studies describe how nurses felt added work pressure due to understaffing [25,26,30,31]. Participants described how they had to assume other responsibilities due to COVID restrictions [34]. Some also stated how they felt that more and more were demanded of them at work [26]. With many participants being reassigned to other departments [26] and other forced work reorganizations it has resulted in higher time pressure [32]. Some nurses stated how they felt overloaded and tired [36]. A negative correlation between intense physical activity during the working hours and sleep length has been found. The correlation implies that an intense physical activity during work hours leads to a decrease of the hours spent sleeping causing healthcare workers to experience drowsiness during the daily work where they had to make significant effort to remain awake [37]. Several participants described having trouble falling asleep [26,27,31] and attributed this to lack of time to decompress mentally despite being physically exhausted [31]. Many also stressed that they were so exhausted and tired that they lacked energy to do anything else [25]. On the other hand, in a study by Diomidous [37] more than half of the participants stated that they did not have any problem with going to sleep. 54.8% stated that they had to get up several times during the night but that they were able to get back to sleep afterwards. One third of the participants stated that they did not get enough hours of sleep.

### 3.3.2. Determinant Sphere: B: Economics and Financial Incentives

(5)     financial situation:

Many of the articles included in this systematic review stated that the nurses worked with limited resources [25,26,28,30,31,34,36,38]. Having insufficient medical supplies such as ventilators and commonly used sedation as well as a lack of protective equipment for instance face shields or masks led to a situation where nurses had to quickly adapt, both by using old or dissimilar mechanical ventilators and no longer being able to rely on having access to supplies and resources [24,25,38]. To some, the lack of PPE indicated a lack of organizational support [31]. The lack of PPE often necessitated a re-use of medical equipment [25,26,36] which caused concern about the diminished effectiveness and protection from COVID-19 infection [26,36].

Nurses stated that they sometimes were unable to provide optimal care for patients due to pressures to reduce costs [35], other nurses felt frustration since, even though there was a lack of manpower, managers refused to increase staff since they claimed they were not making budget [26].

### 3.3.3. Determinant Sphere: C: Relationships, Social Support, and Participation

(6) personal social environment outside work:

A majority of respondents described their fear of spreading COVID-19 to family members [28,30–32,34,38]. Even though proximity to family was described as a need for healthcare workers facing COVID-19 [32] they wanted their families to remain safe. Many of the participants reported taking precautions to protect family members out of fear of infecting them. Most reported taking all necessary precautions at home, isolating from family at home or moving themselves or family members into a different residence temporarily [28]. The fear of infecting others often led to participants distancing themselves from family [31]. The support and appreciation from family and from the public by means of text messages and messages on social media was important as it bolstered resilience [38] and helped stress mitigation [31]. Some nurses felt that they were less available to their families [26]. Unfortunately, some nurses described lack of support and even discrimination from family members [30]. Work-life balance was affected by work schedules [32,33] where prolonged work shifts negatively affecting work-life balance [32]. Additionally, the influence on the scheduling, that rosters only were available at the last minute and lack of organizational logistic affected the nurses work-life balance [32,33].

Some articles described how nurses felt they took work home with them by thinking about the workplace and the patients [25,33], and isolation and living alone were associated with significantly higher levels of depressive symptoms [28]. According to Da Rosa et al., [24] nurses who were divorced, separated, or widowed were more likely to report moderate to severe depression. This was also supported by a study from India [30] were single healthcare professionals had 2.5 times odds of moderate to severe depression and anxiety. However, the results from a study by Galanis et al. [29] showed that nurses who were married, nurses who had children and nurses who lived with others reported higher levels of anxiety. Gago-Valiente et al. [27] found that those who had a partner/were married showed higher emotional exhaustion and high depersonalization.

(7) work social environment at the workplace:

Of the included articles in this review many stated the need for nurses to experience support from others, i.e., mangers and co-workers, in the organisation. Although the organisations were quite effective in developing crisis response policies, providing communication updates, and offering staff emotional support services during the pandemic the need for more effective and appropriate organizational support were reported [24,32,34]. This was also described in a study in Sweden [25] where nurses stated that they lacked support from ICU management. This meant that the nurses themselves were obliged to make decisions about prioritizing nursing and medical intervention. Another study in Brazil [36] where around 50% of the participants found that the support from management was bad or medium at best, seemed to confirm that there globally was a feeling of lack of support. Increased support from the organizations is imperative to reduce nurses' emotional distress and for their physical and mental health well-being [24,28].

Keeping the social distance made it harder to maintain good team climate [32]. Frequently changing teams caused frustration leading to a diminished level of connectedness and trust. However, most participants felt support from colleagues [33]. What was usually a traditional workforce hierarchy was changed to an 'all hands on deck' or 'all in the same boat' approach with nurses describing a newfound inter-professional cooperation and a positive effect on interpersonal relationships within workgroups [31,32] and an increased teamwork [25,36].

Some respondents stated that felt like they had lost their ability to advocate for their individual patient's needs. Some stated that they did not want to speak up about concerns due to the fear of retaliation [34]. Some even described feeling unsafe or bullied by colleagues [35] as well as feeling discriminated against by co-workers [30]. In a study by Thomas et al., [34] some nurses expressed the lack of professionalism from colleagues.

### 3.3.4. Determinant Sphere: D: Execution of Tasks and Activities Relate to Individual and Instrumental Support

(8)    stimulation, appreciation, and motivation within work tasks:

Many articles stated a new way to work and decreased quality of patient care. Several nurses reported both direct and indirect changes in their roles and duties [25,26] where some described that they felt more like medical assistants [25] and that these added duties increased demands on their time, taking them away from the bedside, and diverting them from providing direct patient care [26].

When some nurses were transferred from their wards, the nurses remaining on the ward were left to deal with increased patient care demands [26]. Although ICU nurses were thankful for the support from nurses from other units, they expressed that their workload grew as they continuously had to introduce and help new colleagues. Another aspect that added to their workload was an organisational change that meant more patients per nurse [25].

An adaption to their workflow was common among nurses where they clustered or grouped care tasks in order to preserve PPE and limit the risk of infection while caring for patients who were COVID-19 positive or under evaluation for the infection, even so that it became policy in some wards [26].

During the COVID-19 pandemic patient safety and quality of care were compromised according to nurses who stated that care during the pandemic just had to be 'good enough'. However, not being able to deliver the same standards of care was challenging for the nurses [25]. The need to don PPE and other changes in work processes required more time for preparation and consequently less time providing direct patient care [26]. Respondents also stated that wearing PPE hindered their ability to communicate and form relationships with their patients [25,26,32,33]. Some nurses expressed how patients in the ICU became dehumanized since so little was known about each patient's life and history [25]. Studies described how nurses felt that the enforced safety measures meant that they sometimes had to communicate in less humane ways with patients' family members, i.e., giving bad news over the phone instead of face to face [32,33] and that it was very difficult to give an emotional response by telephone such as putting an arm around them or looking them in the eyes to check how they handled the given information [33].

When it came to nursing care in the ICUs the nurses expressed that there was a feeling of working at an assembly line, since the patients where basically given the same kind of treatment [25]. Some participants expressed frustration due to the actions/behaviour and opinions of the public [34].

However, not all nurses felt hindered by COVID-19 restrictions. In some instances, respondents mentioned that visitor restrictions, which resulted in fewer external support people present in a labor room, lessened distractions and improved their ability to focus on patient-centred care [26].

(9)    competence, skills, and possibility to knowledge development in work

Many articles stated that the nurses experienced that they needed to provide care for patients above their own competence [25,31,35,38]. During the COVID-19 pandemic nurses from other units were transferred to special COVID units including ICUs. In a Swedish study [25] nurses described that they did not receive any introduction, only 19% of non-ICU nurses received an introduction when starting their shifts in COVID-19 ICUs.

Numerous nurses expressed feeling a lack of experience and competence [25,30,31] and that their previous training did not prepare them for working with patients as sick as those with COVID-19 [25,31]. A study from Italy [37] showed that almost half of the respondents were asked to work with more severely ill patients than they were used to. Some nurses described how the management misrepresented the work they were transferred to COVID ICUs to do as they had been assured that they would be to working together with experienced ICU nurses but found themselves often having the sole responsibility for two or more ICU patients [25]. This made them feel like they were forced into a role

that they were not comfortable with, a feeling of caring for patients beyond their expertise, instilling a feeling of being unsupported by management [34]. The importance of teamwork skills such as effective communication, co-operation, and team leadership was important especially since staff needed to be flexible and adaptable while constantly working in different teams with fluctuating levels of competence [25]. Giusino et al., [32] described that some COVID-19 wards were staffed by junior nurses in an environment they were mostly unfamiliar with. A study from the Netherlands [33] found that competence frustration could cause psychological distress. One such frustration came when nurses felt they were required to work with healthcare team members who were not as competent as patient care required [35]. While some studies stated that participants had been given training courses on PPE etc. [29,36,38] some felt that comprehensive training would have been helpful [34]. Some participants described that they have grown professionally due to the pandemic and that they have learnt new skills, especially by communicating with more experienced nurses [25]. One study reported that although some nurses initially felt incompetent to care for COVID-19 patients their confidence was built through training, by using protocols and openly consulting with colleagues [33]. Nurses can experience a high personal fulfilment due to overcoming challenges at work even though showing elevated levels of emotional exhaustion and poor self-perceived general health [27].

## 4. Discussion

The world has gone through the pandemic with severe consequences to, for instance, health and socioeconomic situations. Healthcare workers, and primarily the nurses, were in their work responsible for the care of patients and in the frontline of the COVID-19 pandemic [1–5]. The aim of this systematic review was to identify and analyse the state of knowledge regarding nurses' work situation during the COVID-19 pandemic. Additionally, the health care organizations in most countries address the person-centred care [12–15]. The second aim in this study was therefore to identify any knowledge gap regarding the nurse's work situation with importance to their own health and to a better person-centred health and medical care. The results did identify a knowledge gap regarding person-centred health and medical care since there were not any publications at all detected in the search process of the databases CINAHL, PubMed, Medline, and Scopus. But, to perform a person-centred care the nurses need to experience person-centredness in their own work situation influenced by the work culture of the organization [12–14]. It is an interesting finding that no studies about person-centred care could be found in our search considering the fact that healthcare in more and more countries is evolving into a person-centred healthcare. Therefore, it is of immense importance to develop new knowledge in direction to understand if and how the organization enables the person-centeredness circumstances in the nurses' work situation.

(1) the employees' own self-rated health and diagnoses:

Health and diagnoses

Employee health relates to their work environment, and the employees' health is of significant importance to their ongoing employability and to a sustainable working life [8–11]. The COVID-19 pandemic has had and is still having a negative effect on nurses' health, both physically with decreased skin integrity, muscle, and limb pain, as well as headaches being the most common symptoms [28]. This is further described in a study from Italy [39]. According to one study, the wearing of PPE increases the intensity of the nurses' work by requiring more physical energy, causing hypoxia, and physical symptoms such as fatigue and muscle pain [40]. The pandemic has especially had a psychologically negative effect on the nurses' health. High levels of anxiety, depression, and burnout were described in several studies during COVID-19 pandemic [24,26–32,35,36,38], for instance in the study from the U.S. [28] that stated that both nurses that spent an increased number of hours in close contact with patients infected with COVID-19 as well as healthcare workers that were diagnosed with COVID-19 reported higher levels of depression, anxiety, and burnout. This could be due to the knowledge that they were likely to infect others and knowing that their

safety precautions failed in some way. One previous study shows that there is an association between clinician burnout and suboptimal care practises, medical errors, and decreased teamwork [41]. Several participants in studies from the reviewed articles stated that the fear of getting infected by COVID-19 caused anxiety [24,26,30,32,38]. This is also recognized in other studies [40,42], as well as by the International Council of Nurses (ICN) [43], which states that nurses have always worked under intense psychological pressure but that the current pandemic is making extraordinary demands on nurses both physically and mentally. Both mental and emotional exhaustion were reported by the healthcare professionals in the review's included studies [24,25,27,31]. Mental and emotional exhaustion was also described in other recent studies, such as a social media study [44], where both anxiety due to being a frontline worker during the pandemic and sadness caused by witnessing patients decline and die was described as well as a study from China [45] that showed that 40–45% of frontline nurses experienced anxiety and depression. Feeling anxious and depressed can hinder the nurse's ability to provide person-centred care to the patients they care for. A study states that self-awareness and professional competence are important prerequisites for person-centred care nursing [46]. Sadly, results from a study made in China [40] showed that about 6.5% of respondents reported suicidal ideation. This was also found in a study from the U.S [42] where 5.4% reported suicidal ideation. One study suggests that organizations should implement resource allocation, shared decision making and wellness training programs in order to mitigate stress and burnout [47]. Since we know that stress and burnout are factors that can contribute to a decreased mental health, it is important that organizations stay on high alert to pre-empt healthcare workers' downward spiral into a negative state of mental health by offering support and making sure that the work situation is manageable.

Physical work environment

The physical work environment is about the physical demands, heavy lifting, twist, repetitive and static work, the personal protective equipment, etcetera. The physical work environment is of immense importance for the employee's wellbeing and health, for the possibility to stay employable and to a sustainable work situation [8–11]. The physical work environment left a lot to be wished for, having to work in makeshift ICU wards [25], wards that nurses deemed inadequate and unsafe [36], and wards that lacked proper workspaces as well as a lack of PPE [24–26,31] made caring for patients with COVID-19 more difficult. The need for converting hospital units to specialized COVID-units in order to care for the sheer number of patients that had contracted the virus was also recognised in another study [44]. The work environment was most likely affected by the requirement to wear protective gear as well as keeping a social distance, thus affecting the nurses' ability to communicate with each other. Nurses caring for patients diagnosed with COVID-19 faced an increased risk of getting infected due to proximity and prolonged exposure. Labour and delivery nurses described that women they cared for were not always able to keep their face masks on during labour [26]. A lack of staff [26,30], an increased workload [25,30], wearing personal protective equipment or having to work with patients that were severely ill were factors that affected the nurses' health and well-being and caused a diminished quality of care, let alone a person-centred care. It is known that staff shortages in the healthcare sector have been a fundamental problem in Sweden as well as in other European countries even before the pandemic [48]. One study state that increased workloads lead to understaffing of nurses, preventing them from spending enough time with patients, thus rendering them unable to meet the duty of care [49]. One pre-pandemic study state that the adverse physical working conditions common in the nursing profession is one of the main causes why nurses in the Nordic Countries leave the profession at an early age [50]. We might be able in the short run to keep the healthcare going by educating new nurses, but there will come a breaking point when that very fragile balance tilts in a negative way, therefore it is vital to put effort into how to retain nurses in the healthcare sector. The WHO defines a healthy workplace as one in which workers and managers collaborate to use a continual improvement process in order to protect and promote the health, safety and well-being

of all workers and the sustainability of the workplace [3]. The COVID-19 pandemic is most likely not the last pandemic that we will encounter, and it is important that we learn from our mistakes and experiences and prepare for when, not if, a new pandemic occurs. Hopefully, by making sure that we take what we have learned during the pandemic and ensure that there is enough staff, enough up-to-date equipment, relevant protocols and making sure that we have a plan to ensure access to enough materials and resources, we will be better prepared when the next crisis hits.

Mental work environment

The mental work environment with stress within the work situation, threat and control are of significant importance to the employees' wellbeing and health and to a sustainable work situation [8–11]. Many nurses in the studies described loss of control, whether it was from not being able to influence their schedule [32,33], their working conditions [25] or being transferred to other wards and required to join new teams [26,33] caused stress and affected the nurses' work-life balance. Having to join an unfamiliar team were no one knows your qualification and you do not know the location of the medical equipment, or the wards rules can cause anxiety and leading to the nurse feeling isolated. Having high demands put on them as well as having low control of their work situation can cause a tense work situation for nurses [15]. The uncertainty by not knowing exactly what the disease would bring and not knowing exactly what to expect and what was expected of you as a nurse is sure to have caused frustration. Hence the need for having colleagues to whom one can depend on. Something that could have helped nurses feel more in control is better transparency and open communication from the organization about what was happening in the different wards. During the COVID-19 pandemic nurses have been exposed to highly stressful situations such as coworkers getting infected/admitted to hospital [28], deteriorating patients [33], and even having to witness patients dying [32,34]. This was also reported in a study who states that nurses who experienced high death tallies, having to care for dying patients and caring for the patients' families caused an emotional trauma [51]. This is sure to have caused a mental strain on the nurses. Previous research has shown that repeated exposure to stressful patient-related situations can make nurses especially vulnerable to stress-related outcomes such as emotional exhaustion [52] and that stress-related outcomes can lead to serious consequences including lower productivity, increased risk of medical errors and a higher turnover intention [53].

Several studies [26,34,35,38] indicated that nurses were considering leaving the profession due to a range of factors. This is also what is found in resent literature, according to a study [54] one fifth of the respondents intended to leave their position in the next six months, the ICN [55], states that the lack of protection, and long and stressful shifts are severely impacting nurses' mental health, resulting in nurses leaving or planning to leave the profession. This is also described in a study [56] where they found that the psychological impact of working during the pandemic has resulted in several nurses actively considering leaving their career. Job demand was associated with increased nurse turnover, which implies that to decrease nurse turnover, it is necessary to reduce job demands [57] and to focus the job resources, especially leadership [58]. By having a more supportive and transparent leadership that strives to decrease the stressful work environment for healthcare staff and making sure that there is a good atmosphere at the workplace, chances are that the healthcare organizations can retain qualified staff.

Working hours, work pace, time for recuperation

The employees work schedule, work hours, work pace and time for recuperation are of immense importance in a sustainable work situation and are also especially important to stay employable [8–11]. A correlation was found between intense physical activity during working hours and diminished duration of sleep [37]. A few studies [26,27,31] showed that some participants had trouble falling asleep despite being exhausted. Then again, according to one of the studies made in Greece [37] over half of the participants stated that they did not have any trouble falling asleep. However, one third of participants in the

same study stated that they did not get enough sleep. A study [59] showed that nursing staff in general who cared for patients with COVID-19 had significantly higher rates of insomnia. Both having trouble falling asleep and the diminished duration of sleep could indicate lack of time to decompress mentally but also be indicative of burn-out symptoms. Several studies showed that nurses worked longer hours [25,32,36], experienced a greater workload [25,30–32,35,36], felt added work pressure due to understaffing [25,26,30,31], and felt overloaded and tired [36]. An increase in working hours was also reported in a study which claims that nurses are now forced to work for exceedingly extended periods of time without respite [51]. This seems to indicate that the nurses often do not have enough time to recover, neither mentally nor physically, between shifts. Studies have shown that when staff are tired chances of work-related injuries as well as the risk of doing the wrong thing at work such as medication errors increases [59–61], the prevalence of adverse events were approximately 65% in those who reported excessive daytime sleepiness [61]. One study found that neurobehavioral deficits due to lack of sleep are greater in the younger population [62]. One can ponder whether this might be a reason why the younger generation of nurses are the ones leaving the profession. Workload and lack of emotional support at the workplace were especially predictive of mental distress regarding burnout among nurses during the pandemic [63]. Higher worktime demands were strong predictors of emotional exhaustion [64,65], with low autonomy being the strongest predictor [65]. Being able to take breaks at work as well as to be able to influence other aspects of work has showed a positive effect on work-related fatigue, sleep, and health complaints [66].

During the, so far, almost two-year pandemic many nurses have had an increased workload. One study state that it is known that constant disproportionate workloads among health care professionals are damaging to the quality of patient care, contribute to an increase in workplace mistakes and staff turnover and negatively impact work satisfaction [67]. Having a positive time experience, which is having enough time to do your tasks without time pressure both in the private domain as well as in the work domain, leads to better recovery which in turn is strongly linked to higher subjective health and increased quality of life [68]. We need to consider that in order for nurses to have more time with patients and to be able to improve the quality of care they need a feasible workload. Therefore, more nurses are needed in the workplace. By adding more staff and managing workloads, the organizations would promote health and well-being among the co-workers, thus creating a sustainable workplace.

Financial situation

Work is a possibility to finance individuals living, and sick leave, the risk of unemployment and to stay employable are of significant importance in relation to a sustainable work situation [8–11]. The pandemic has forced nurses and other healthcare workers to work with limited resources such as having to work with other sedatives than the preferred one for patients on mechanical ventilation, work with ventilators that have not been used for years and a lack of PPE [24–26,31], causing the need for re-use of medical equipment [25,26]. Not being able to rely on access to supplies and resources [24,25,28] caused extra stress and could also increase the risk of being infected, on sick leave, or seriously ill. The financial situation in the organisation effects the work environment. One study reported that over half of the respondents had difficulty accessing PPE and were therefore forced to reuse or extend the use of masks and face shields [49]. The lack of necessary clinical equipment and protective gear was a hindrance for safe care delivery in general during the pandemic [51]. By using disinfectant on for instance face shields concerns were raised with the diminished effectiveness of the material [26]. The lack of PPE was probably caused by different reasons, in Sweden one of the factors that caused shortage of PPE was not being enough prepared for what the pandemic would entail as well as the duration of the pandemic whilst in other countries, one can speculate, the lack can be due to limited funds. Nurses felt that pressures to reduce costs [35] affected their work. Having easy access to adequate equipment/resources has previously been described as resources for person-centred nursing [69]. Having to work with limited resources during the pandemic is nothing we can change at

the moment; however, it is important to make sure that when and if a comparable situation arises, we are better prepared thus also making sure that the patients receive quality care.

Personal social environment

The private life has importance to the working life and effect if the employee can and want to stay in the workplace and to stay employable [8–11]. Fear of spreading the virus from work to family members or vice versa was something that almost all studies touched on [28,30–32,34,38]. This is understandable since COVID-19 was something that was not quite known in regard to route of infection, possibilities of treatment etc.

Nurses described that they felt like they took their work home with them [25,33], something that is not surprising considering the vast effect the pandemic has had on the healthcare, healthcare workers and the world in general. According to a study from Turkey [70] it is suggested that nurses take less time for themselves and focus more on their work life thus negatively affecting the work-life balance. The restrictions put in place during the COVID-19 pandemic may be a relevant factor to decreased work-life balance since restaurants, gyms etc. has had limited operating hours. Another study [71] showed that low levels of work-life balance was associated with higher levels of intention to leave the profession.

The importance of support from family and the general public was evident in the reviewed articles as was the concern from healthcare workers of infecting family with COVID-19 virus. Some participants stated that they distanced themselves from family members to keep them safe, this separation can increase feelings of anxiety and depression [31]. Something that was interesting was that some studies found that nurses who were single, divorced, separated, or widowed were more likely to report moderate to severe depression and anxiety [24,30]. In contrast, the study by Galanis et al. [29] showed that nurses who were married, nurses who had children and nurses who lived with others reported higher levels of anxiety. According to the study made in Spain those who had a partner/were married showed higher emotional exhaustion and high depersonalization [27]. One can speculate whether or not the social support surrounding nurses influenced their emotional health. A study from China shows a significant correlation between social factors and nurses' psychological well-being [40]. A perceived lack of support from family and/or organization was shown to be important risk factors against poor mental health. Too little attention is paid to social wellbeing even though health is not only the absence of disease or injury but so many more aspects [72]. It is important to achieve a good balance between work life and personal life. Having support from family and friends as well as being able to leave work behind at the end of the day is vital to one's physical and mental health.

Work social environment

The social work situation, the leadership, trust, and participation in the work group are of immense importance in a sustainable work situation [8–11]. A wish for increased support from the organization was expressed in several studies [24,25,32,34,36]. Participants in one study stated that nurses had decreased Quality of Life and suffered from increased moral distress due to the lack of awareness and support from executive leadership [73]. Another study showed that trust and connectedness can decrease in teams that frequently has changing team members which could cause frustration [33]. However, articles included in this review stated that the sense of teamwork seemed to increase during the pandemic [31,32,36]. This was also found in a study where a majority of HCPs reported an improved inter- and intradisciplinary collaboration as well as what was expressed by some respondents as a newfound sense of respect for other members of staff [73]. Something that was disconcerting was that some nurses avoided speaking up about their concerns out of fear of retaliation [34], and that they felt unsafe and experienced bullying [35] and/or discrimination [30] from co-workers. This is something that is unacceptable in the workplace and needs to be delt with. Employees in a study reported that good emotional health comes from working in a respectful workplace and that mental and physical well-being is a prerequisite for being able to cope with the demands of the job [74]. Having positive social

relationships at work can alleviate the burden of both emotional demands and worktime demands [75]. Colleague belongingness enhances wellbeing as it satisfies a human need to be confirmed and to belong [76]. Colleague belongingness consisting of trusting, supportive, and a positive work environment was a valuable resource for employee health [77]. Relationships are important for employees growing and flourishing. Therefore, in order to create sustainable workplaces as well as organizations with flourishing individuals, it is important that the work structure is built on positive relationships. Healthy relationships among employees can also promote a person-centred culture [78]. The organizational culture, attitudes, values, and beliefs affect the mental and physical wellbeing of employees [3]. The organisational culture of a workplace including structure and communication is what makes a workplace a health promoting workplace [74]. According to a study made in Sweden, communication is a key component for professional competence and is related to a holistic, person-centred nursing approach [69]. Therefore, it is important to make sure that communication is prioritized, especially during a pandemic when there is a lot of information and directives that needs to be distributed.

Stimulation, appreciation, and motivation

To experience stimulation and appreciation in the tasks and the work situation is important to the work motivation and of significant importance to want to stay employable in the workplace [8–11]. Changes in roles and responsibilities [25,26], a decreased quality of care and not having enough time by the bedside [26] were all factors that the nurses described their work situation to be like during the COVID-19 pandemic. ICU nurses described how their workload grew due to continuously having to introduce and help new colleagues [25].

The increased demands on nurses' time which took them away from the bedside [26] also affected the nurses' possibilities of providing person-centred care. Another factor that affected the ability to provide person-centred care was the need to wear PPE which hindered nurses' ability to communicate properly and form connections [25,26,32,33] with the patients. Having to give information/news over the phone [32,33] could cause moral stress for healthcare staff, especially since they could not see how the news/information was received. When news was given face to face, they could put a comforting arm around the shoulder or in other ways offer support, something that was limited during the pandemic. This was also described in a study who found that sharing life-altering diagnoses or a poor prognosis virtually added a layer of complexity to difficult conversations as it eliminated the 'caring touch' [79]. Nurses in the ICU stated that patients became dehumanized and were basically given the same treatment [25] which is not in line with person-centred practice where the patient is the centre. Those who found meaningfulness in their work reported better health [77]. This makes sense if we for instance consider sense of coherence, which is a way for people to cope with everyday life stressors and that consists of three core components: comprehensibility, manageability, and meaningfulness [80]. None of the participants in the studies in this review described feeling appreciated by management, the appreciation came from the public and in some degree from colleagues. Previous studies have found that rewards and appreciation is a strong predictor of nurses' job satisfaction and work engagement [81], both factors that are known to contribute to nurse turnover [82,83]. One study state that work tasks can be perceived as negative when there is a discrepancy between how motivated the employee feels and to what extent that employee feels appreciated [84]. Feeling appreciated and that the work you do matters is particularly important, especially during strenuous times such as in a pandemic and it does not take a lot of time or effort from organizations to express this to the healthcare staff.

Competence, skills, and possibility to knowledge development

Knowledge and competence on how to manage the tasks are of immense importance in a sustainable work situation, and the right and enough knowledge, skills and competence are also particularly important to stay employable [8–11]. Around the world the COVID-19 pandemic caused a significant volume of acutely or critically ill patients. Nurses were

redeployed to other wards, nursing students were asked to help out in the wards, and retired nurses were asked to help with vaccinations. However, nurses described how they were given little or no introduction [25] when they were transferred to other wards and some COVID wards were staffed with junior nurses that were mostly unfamiliar with the environment [32] and/or the severity of the patients. In many countries hospitals cancelled elective surgeries and closed certain wards and outpatient clinics in order to reallocate staff to emergency departments, ICU wards, COVID- units etc. Concerns for the lack of experience and competence for working with acutely ill patients were raised by several nurses [25,30,31]. Having to work with patients beyond their expertise and knowledge [25,31,35,37,38] made nurses feel anxious and unsupported by management, often causing moral distress. Exposure to events that caused moral distress was more common during this pandemic and was strongly correlated with mental illnesses such as depression and PTSD. The frequent occurrence of moral distress was due to the intensity of the working environment, the exposure to death and the changes to the work environment that has led to healthcare staff working in unfamiliar conditions [56].

The COVID-19 pandemic can lead to personal development since so many nurses in the world faced a challenge in their profession, forcing them to learn new things. Nurses felt that they have grown professionally [25,33], which is important especially since we know that many nurses are leaving the profession. If nurses who are beginning their careers can meet nurses who have excelled and flourished in their profession perhaps they would be more inclined to remain in the profession.

Limitations

Something that could affect the results in our study were the chosen keywords, perhaps if we had used other keywords or other databases the result would be different. Some of the articles had snowball sampling which can be debatable whether or not this affects the validity. When considering internal validity, it is unclear whether the respondents referred the researchers to other colleagues that shared their opinions, on the other hand, the researched population consists of nurses that have faced similar situations in the COVID-19 pandemic meaning that the external validity and transferability was strengthened.

The second aim in this study was to identify any knowledge gap regarding the nurse's work situation with importance to their own health and to a better person-centred health and medical care. It can be perceived as a shortcoming that no article was detected in the search process of the databases CINAHL, PubMed, Medline, and Scopus, when person-centred was added to other keywords. However, this result also identifies a possible knowledge gap in the direction to a better person-centred health and medical care. It is interesting that no study was found when healthcare in more and more countries is evolving into a person-centred healthcare and that probably had to be especially central in the COVID-19 care and effect the nurses work situation. This is an important finding, and therefore more research is needed to increase knowledge in this area in the future. Though, by using the definition words of person-centred care in the analysis of the articles we could identify several crucial factors in the nurses' work environment that influenced their ability to practice person-centred care. Not being able to/or forgetting to work in accordance with person-centred care, especially during a pandemic when it is of the utmost importance, is a great loss to the quality of care provided to patients as well as a potential source of moral distress to nurses.

A strength of this study is that, to our knowledge, no other review article has looked at nurses' work situation during the COVID-19 pandemic so thoroughly i.e., through all nine determinate areas of the swAge model. Another strength is that both researchers CN and KN were diligent in the research methodology and included all relevant articles.

The study populations in the included articles were primarily nurses but other health-care professionals such as medical doctors, EMTs and nursing assistants were included as well. Both men and women from the ages of 18 and up and from various parts of the world (U.S, Sweden, Greece, Japan, Korea, Turkey, Italy, Spain, Brazil, India, and the Netherlands) participated. By including studies from several parts of the world, that also describes pretty

much the same experiences of nurses and where the same problems and possibilities were identified despite the fact that the studies were performed in different parts of the world, can be considered a strength since the transferability is very plausible. Additionally, a majority of determinate areas from the swAge model is addressed by at least two thirds of the included articles. Determinate area 2 (Physical work environment) were only addressed by 7 articles and determinate area 8 (Motivation, stimulance, task satisfaction) were only addressed by 5 articles so these areas could be considered as underrepresented. However, one can speculate that these are areas that need further investigation.

## 5. Conclusions

One of the most important results in this review is that there was no research available that explicitly describes person-centred care in the nurses' work situation during COVID-19. The articles were analysed through the determinate areas of the swAge model to identify factors of importance to the nurses' sustainable work situation and a healthy working life that contributes to long-term employability. We found that all the determinate areas are of importance to both the nurses' sustainable work situation during the COVID-19 pandemic and to person-centred care. Content related to *Diagnosis and self-rated health*, *Mental work environment* and *Private-social environment* was, however, most frequently identified through the analysis of the articles. There is a further need for more studies that addresses person-centredness from an organisational perspective with the intention to develop strategies and measure activities on how to make the nurses work situation more sustainable, and to increase their possibility to perform a more person-centred care even in times of a pandemic. There is also a need for more research to effectively promote healthcare workforce well-being as well as an increased understanding of what can entice the nurses to stay in their profession and what can be done to make nursing a more attractive profession. Many countries experienced a lack of healthcare workers prior to the pandemic and since numerous studies have indicated that many nurses are considering leaving the profession it is vital that we try to both educate new healthcare workers and most importantly try to retain the healthcare workers that we do have.

**Author Contributions:** C.N. and K.N. conceptualization, methodology, validation, formal analysis, writing—original draft preparation, writing—review and editing. A.W., Å.B., S.S.P. and P.N.L.—review and editing. All authors have read and agreed to the published version of the manuscript.

**Funding:** Funding was provided by the Research Platform for Collaboration for Health, Kristianstad University, Sweden. Funding number 9/2021.

**Institutional Review Board Statement:** Not applicable.

**Informed Consent Statement:** Not applicable.

**Data Availability Statement:** Not applicable.

**Conflicts of Interest:** The authors declare no conflict of interest. The funders had no role in the design of the study; in the collection, analyses, or interpretation of data; in the writing of the manuscript, or in the decision to publish the results.

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
