# Peer review of "Nurses’ Work Environment during the COVID-19 Pandemic in a Person-Centred Practice—A Systematic Review"

_sustainability, doi:10.3390/su14105785_

Round 1
Reviewer 1 Report
Dear authors,
The objectives of the systematic review are very interesting, topical, and of scientific interest for the context of nursing professionals during the Covid-19 pandemic.
However, regarding the methodological design, you have indicated that it is a systematic review. There is extensive literature on how to carry out a systematic review, which supports the need to implement very specific steps. These include assessing the quality of the selected studies.
The study generally follows these methodological phases, except for the assessment of the quality of the studies they have selected, and the process of selecting the final sample. In this sense, it is also necessary to specify how many reviewers conducted the search independently and how the results were pooled in order to reach a consensus on the final sample of articles selected.
I encourage you to extensively review the literature on the systematic review method, and to revise the presentation of your results in this manuscript, bringing them into line with this methodology. If, on the other hand, you did not conduct a systematic review, as there are many types of reviews, then you should change the description of the study design.
Thank you very much for the data you provide, I think they are interesting for the design of strategies to improve psychosocial factors related to nursing work.
Yours faithfully
Author Response
Hello, thank you for taking the time to review our article.
We will look into the method section and try to elaborate on the points that you have made.
We have made the following changes:
• Was an assessment of the quality of the studies they have selected made? – Yes, an assessment was made of the quality of the studies by researcher CN and KN. All selected articles were of good quality. We have added text in the methods section to make this clearer.
• How many reviewers conducted the search independently and how were the results pooled in order to reach a consensus on the final sample of articles selected? – Researcher CN and KN performed a literature search in the 4 databases using the mentioned keywords. 257 articles were uploaded into Rayyan database to exclude doublets and to sort by relevance. 52 articles were chosen initially to look at abstracts, method, results and conclusions, this was done by researcher CN and KN individually and marked whether relevant or not in the Rayyan database. We found 17 potentially relevant articles to read in fulltext, this was also done separately by CN and KN where 2 more articles were excluded. The final 15 articles were read by CN and KN and a content analysis was made. We have added text in the methods section to make this
clearer.
Kind regards,
Cicilia Nagel
Reviewer 2 Report
Dear Authors,
I was pleased to read your article on the work environment for nurses during the COVID-19 pandemic. This is a very important topic. The pandemic came as a great surprise to health care systems around the world. It caused reorganization of work and required a lot of financial resources. In addition, health systems struggled with staff shortages. Your work is an important voice in the discussion. I think that the conclusions can be used by many policy makers as practical implications.
I make no comments on the paper. I rate it highly.
Author Response
Hello, thank you for your time and kind words.
Kind regards,
Cicilia Nagel
Reviewer 3 Report
I would like to congratulate the authors for their interest in researching in this field, however, the work presented presents some deficiencies.
a) Abstract. Remove the term background from the section.
b) Abstract is correct in general terms but does not provide sufficient information of the results of the systematic review. What results have been reached based on the 9 points studied?
Authors should include in a summarized form, this information in this section.
c) Keywords should be revised. “Organization” term appears 2 times. Work environment is no representative. Keywords should be reviewed keeping in mind that their job is to help readers identify the field and subject of the article. This is especially relevant in a systematic review.
d) I have missed the numbering of each line of the article. This point is a Sustainability formatting requirement that helps to indicate points and references to your document and should be included in this article.
e) Conclusions section. There are several sub-sections within this section that are difficult to locate (Health and diagnoses, Physical work environment,...) The authors should highlight these sections to help the understanding of the document.
I hope that these changes will help to improve your article and make it a document of great scientific interest.
Author Response
Hello, thank you very much for reviewing our article. I will respond by using the same letters as you.
a) We will remove the term background
b) We will re-write and include more results in the abstract
c) We used organization/organisation since we did not want to miss studies made either in the UK or the US
d) according to my files the numbering is present in the article that I sent in for review, but I will make sure that it is there when I re-submit
e) We will try to highlight the important sub-sections.
Again, thank you for your time and for your comments. I am sure that the changes will improve our article.
Kind Regards,
Cicilia Nagel
Round 2
Reviewer 1 Report
Dear authors,
Thank you very much for the improvements you made.
Yours sincerely,